# In Vivo and In Vitro Pro-Fibrotic Response of Lung-Resident Mesenchymal Stem Cells from Patients with Idiopathic Pulmonary Fibrosis

**DOI:** 10.3390/cells13020160

**Published:** 2024-01-16

**Authors:** Gabriel Escarrer-Garau, Aina Martín-Medina, Joan Truyols-Vives, Cristina Gómez-Bellvert, Linda Elowsson, Gunilla Westergren-Thorsson, Maria Molina-Molina, Josep Mercader-Barceló, Ernest Sala-Llinàs

**Affiliations:** 1MolONE Research Group, University of the Balearic Islands (UIB), 07122 Palma, Spain; 2iRESPIRE Research Group, Health Research Institute of the Balearic Islands (IdISBa), 07120 Palma, Spain; 3Pathological Anatomy Service, Son Espases University Hospital, 07120 Palma, Spain; 4Lung Biology, Department of Experimental Medical Science, Lund University, 08908 Lund, Sweden; 5ILD Unit, Respiratory Department, University Hospital of Bellvitge-Bellvitge Biomedical Research Institute (IDIBELL), 08908 Hospitalet de Llobregat, Barcelona, Spain; 6Centre of Biomedical Research Network in Respiratory Diseases (CIBERES), 28029 Madrid, Spain

**Keywords:** idiopathic pulmonary fibrosis, lung-resident mesenchymal stem cell, transforming growth factor β, bleomycin, inflammation, extracellular matrix proteins, myofibroblast

## Abstract

Lung-resident mesenchymal stem cells (LR-MSC) are thought to participate in idiopathic pulmonary fibrosis (IPF) by differentiating into myofibroblasts. On the other hand, LR-MSC in IPF patients present senescence-related features. It is unclear how they respond to a profibrotic environment. Here, we investigated the profibrotic response of LR-MSC isolated from IPF and control (CON) patients. LR-MSC were inoculated in mice 48 h after bleomycin (BLM) instillation to analyze their contribution to lung damage. In vitro, LR-MSC were exposed to TGFβ. Mice inoculated with IPF LR-MSC exhibited worse maintenance of their body weight. The instillation of either IPF or CON LR-MSC sustained BLM-induced histological lung damage, bronchoalveolar lavage fluid cell count, and the expression of the myofibroblast marker, extracellular matrix (ECM) proteins, and proinflammatory cytokines in the lungs. In vitro, IPF LR-MSC displayed higher basal protein levels of aSMA and fibronectin than CON LR-MSC. However, the TGFβ response in the expression of TGFβ, aSMA, and ECM genes was attenuated in IPF LR-MSC. In conclusion, IPF LR-MSC have acquired myofibroblastic features, but their capacity to further respond to profibrotic stimuli seems to be attenuated. In an advanced stage of the disease, LR-MSC may participate in disease progression owing to their limited ability to repair epithelial damage.

## 1. Introduction

Idiopathic pulmonary fibrosis (IPF) is a progressive disease with no curative treatment. It is an aging-related and heterogeneous disease [1,2] caused by an abnormal response to damage in which transforming growth factor β (TGFβ) and other signaling pathways involved in tissue repair are dysregulated [3]. Such dysregulation involves the loss of alveolar epithelial cell identity, the proliferation of fibroblasts, myofibroblast differentiation, and the excessive production of extracellular matrix (ECM) proteins [4]. This lung tissue remodeling is irreversible and implies a progressive loss of respiratory function.

Lung-resident mesenchymal stem cells (LR-MSC) execute a pivotal function in tissue repair and regeneration in healthy conditions [5]. However, in IPF, the involvement of LR-MSC in this aberrant and dysregulated repair is not fully understood due to the limited number of studies [6]. The first evidence of the role of LR-MSC in pulmonary fibrosis came from the bleomycin (BLM)-induced fibrosis animal model, in which it was demonstrated that LR-MSC were depleted after BLM administration [7] since their proliferation and differentiation into myofibroblasts [8]. Specifically, murine LR-MSC can differentiate into myofibroblasts through the activation of hedgehog and Wnt/B-catenin signaling and sumo-specific protein 1 (SENP1)-mediated deSUMOylation [9,10,11]. Thus, the transformation of LR-MSC into myofibroblasts suggests that they actively contribute to pulmonary fibrosis activity [12,13,14,15]. Moreover, other studies also conducted in the BLM model identified progenitor cell populations as a significant source of ECM-producing cells that express the myofibroblast marker α smooth muscle actin (aSMA) and expand after BLM injury [16,17]. The BLM animal model of pulmonary fibrosis presents, however, several limitations for the study of IPF [18]. In contrast to IPF, mice treated with a single BLM dose resolve lung damage, and, therefore, this is a key issue to consider in the study of the role of LR-MSC in IPF. Importantly, LR-MSC from IPF patients are expected to be continuously exposed to profibrotic stimuli, whereas in BLM-treated mice, profibrotic stimuli decrease, and such a change may affect LR-MSC behavior. Therefore, studies using human LR-MSC are required to understand the role of these cells in the evolution of IPF.

In humans, the number of lung perivascular ABCG2pos MSC isolated from IPF patients decreased [8], in agreement with several findings shown in experimental models. We recently demonstrated that IPF LR-MSC showed a lower expression of genes involved in oxidative phosphorylation and mitochondrial dynamics than control LR-MSC, a limited respiratory capacity, and dysmorphic mitochondria [19]. Accordingly, the limited repair potential of IPF LR-MSC seems to be, in part, associated with stem cell exhaustion and mitochondrial dysfunction. Moreover, recent data also suggested that LR-MSC could actively contribute to fibrosis development in IPF [20,21]. The pathologic role of fibroblasts is acquired at the early stages of lung mesenchymal progenitor cell differentiation [20], and the expression of ECM-related genes is increased in all mesenchymal subpopulations [21]. The pathologic role of LR-MSC seems to be influenced by their microenvironment [22,23]. Thus, IPF LR-MSC displayed a high expression of genes involved in inflammation, oxidative stress, and hypoxia [23]. Interestingly, Bonifazi M et al. [23] also showed that IPF LR-MSC induced the expression of some pro-inflammatory cytokines, TGFβ, COLA1A, and αSMA, in co-culture with control MSC or fibroblasts, indicating that IPF LR-MSC induces, in vitro, a pathological phenotype in the surrounding cells. Conversely, the lung microenvironment is critical for MSC capabilities [22]. Most of these findings were described in isolated LR-MSC and progenitor cells cultured under basal conditions. However, it is unclear how these cells behave in a profibrotic milieu.

In the present study, to better understand how they behave in a profibrotic environment, we analyzed the response of LR-MSC from patients with IPF compared with those isolated from controls to profibrotic stimuli, both in vivo, using an experimental model of BLM, and in vitro, exposing LR-MSC to TGFβ.

## 2. Materials and Methods

### 2.1. LR-MSC Isolation, Validation, and Culture

The animal procedure was approved by the Ethics Committee for Animal Experimentation of the Balearic Islands (2019/09/AEXP). The collection of human LR-MSC was approved by the Balearic Islands (IB 1991/13 PI; IB 3335/16 PI) and Swedish (2008/413 (2022-01221-02), 657-12 (13 September 2012, 2006/91) Ethical Committees. LR-MSC were obtained from the reservoirs of 3 independent cohorts: Hospital Universitari Son Espases (Palma de Mallorca, Spain), Hospital Universitari Bellvitge (Barcelona, Spain), and Lund University Hospital (Lund, Sweeden), and designated either to IPF or control (CON) groups. IPF patients were included in this study according to the guidelines followed for IPF diagnosis based on ATS and ERS criteria [24]. IPF patients from the Spanish cohort presented a mild/moderate degree of disease, whereas IPF patients from the Swedish cohort presented a severe degree.

LR-MSC were isolated and identified as described in [19]. Briefly, lung tissue of IPF patients was obtained from biopsies or explanted tissue, whereas for control individuals, lung tissue came from postmortem persons or from the surrounding tissue of tumor resections. Tissue samples were processed to isolate LR-MSC using standardized protocols [19,25]. The obtention of MSC was validated using the BD StemflowTM human MSC analysis kit (BD Biosciences, 562245, Piscataway, NJ, USA), the human mesenchymal stem cell functional identification kit (R&D Systems, SC006, Minneapolis, MN, USA), and the Rohart test [26], or immunophenotyping and tri-lineage differentiation, as it is established in [27]. Cells were cultured for 4–6 days in αMEM medium (Biowest, L0475-500, Nuaillé, France) supplemented with 10% FCS (Biowest, S1810-500), and 1% penicillin/streptomycin (Capricorn Scientific, PS-B, Ebsdorfergrund, Germany) inside a cell incubator at 37 °C, 5% CO_2_, and 98% humidity. Non-adhered cells were removed by changing the media, and the adhered cells were kept in culture up to passage 6–7 prior to the administration to mice.

### 2.2. Pulmonary Fibrosis Induction, LR-MSC Instillation, and Sacrifice

Three-month-old female C57BL/6 mice were used in this study. Mice (*n* = 35) were divided into two groups according to their body weight (PBS, 20.3 g, *n* = 15; BLM, 20.2 g, *n* = 20), anesthetized with ketamine/xylazine, and i.t. instilled with a 75 μL sterile solution of PBS or 2 U/kg of BLM (day 0). To analyze the effect of LR-MSC exposed to a profibrotic environment on fibrosis evolution, LR-MSC were administered 48 h after BLM instillation. Thus, mice from PBS and BLM groups were divided into three subgroups of 5–7 mice each, depending on whether they received vehicle, CON, or IPF LR-MSC. A total of 6 experimental groups were formed, as depicted in Figure 1A. LR-MSC were administered by a veterinarian with practical expertise in i.t. administrations. Mice were anesthetized and i.t. instilled with a 100 μL solution containing 100,000 cells from a pool of three different male IPF or CON patients, or the same volume of sterile PBS. Cell suspensions were mixed by inversion, and immediately connected to the cannula. To have a representative pool of IPF LR-MSC, two patient samples from the Spanish cohort (mild/moderate) and one patient sample from the Swedish cohort (severe degree) were selected. Health status was checked daily, and body weight was registered on days 0, 2, 4, 8, 10, and 14.

On day 14, mice were euthanized in a CO_2_ chamber. A tracheotomy was performed to collect the bronchoalveolar lavage fluid (BALF) by flushing twice with 500 μL of sterile PBS with a cannula. Lungs were washed through the heart with sterile PBS, and the right lung was ligated. All right lung lobes were snap-frozen for molecular analysis, and the left lung was fixed in paraformaldehyde for histological analysis.

### 2.3. Histological Analysis

Lung tissues were fixed in 10% formalin and embedded in paraffin for histological examination. Two 4 μm thick lung sections at a 25 μm depth were obtained for Masson’s trichrome staining, and one 5 μm thick section was stained with Picrosirius Red. Picrosirius Red staining was performed as detailed in [28]. Briefly, deparaffinized and rehydrated lung sections were immersed in a 1% phosphomolybdic acid solution for 2 min, rinsed in water, incubated in a saturated picric acid solution containing 0.1% Direct red 80 (Sigma-Aldrich, Burlington, NJ, USA) for 2 h, and finally washed in 0.01 N HCl, dehydrated, and fixed. Tissue sections were digitalized by taking and fusing files at 50× magnification using a D1 Cell Observer Zeiss. Lung fibrosis was evaluated from the lung sections stained with Masson’s trichrome by a pathologist in a blind manner using the modified Ashcroft scale for small animals [29] in 10 lung consecutive fields per animal. Collagen quantification in picrosirius red-stained samples was assessed by image analysis in accordance with [30], scanning the entire lung sections. Images were processed using image J software (version 2.3.0/1.53q, National Institutes of Health, Bethesda, MD, USA). The entire lung sections were manually defined, and the total lung area was calculated. A green channel image was generated by splitting red, green, and blue channels. Total collagen was quantified after removing background noise, and data were reported relative to the total lung area. Representative images of Masson’s trichrome and picrosirius red staining obtained with the 3DHistech P1000 digital scanner are shown in Appendix A, respectively.

### 2.4. BALF Cell Count

BALF cells were obtained after centrifugation at 800× *g*. A total of 20 μL of cell pellet was mixed with 180 μL of 1/10 diluted lysis buffer solution (BD Biosciences, San Diego, CA, USA) and incubated at room temperature for 10 min. Samples were analyzed using a BD FACSVerse cytometer.

### 2.5. mRNA Expression

Total RNA was isolated from lung samples and cultured LR-MSC using TRItidy G™ (PanReac AppliChem, Barcelona, Spain) following the manufacturer’s protocol. 150 ng of RNA was subjected to cDNA synthesis with the Transcriptme RNA kit (Blirt, RT31, Gdansk, Poland). Finally, cDNA was used for PCR analysis using the SensiIFAST™ SYBR^®^ No ROX Kit (Bioline, BIO-98005, Antipolo City, Philippines) and the primer sequences detailed in Appendix A.

### 2.6. Protein Expression

Lung tissues were homogenized to powder and lysed in RIPA buffer. Protein was quantified by Bradford assay (Thermofisher, #23227, Waltham, MA, USA), and 30 µg of protein was boiled in loading buffer 4× for 5 min for Western blotting. Samples were loaded in 10% acrylamide gels, and running was performed in a running buffer at 120 W using a Mini-TransBlot device (Bio-Rad, Hercules, CA, USA). Semi-dry transfer to PVDF membranes (Millipore, Burlington, NJ, USA) was performed by the Trans-Blot Turbo system (Biorad). Membranes were incubated in blocking buffer (5% BSA, 2% tween in tris-buffered saline (T-TBS)) for 1 h at room temperature, followed by incubation with primary antibodies (diluted 1:1000 in blocking buffer) overnight at 4 °C. Membranes were washed three times in T-TBS and further incubated with HPRT-conjugated secondary antibodies (diluted 1:2000 in blocking buffer) for 1 h at room temperature. Development was done by chemiluminescence (Supersignal West Dura substrate, Thermofisher) using the ImageQuant™ LAS 4000 system (GE Healthcare Bio-Sciences AB, Uppsala, Sweden). Protein levels were quantified with ImageJ using bActin (A3854, Sigma, Burlington, NJ, USA) for loading control. Primary antibodies used were to detect cytokeratin 8 (KRT8) (ab5328, Abcam, Cambridge, UK), aSMA (A5228, Sigma, Burlington, NJ, USA), and KRT18 (04-586, Millipore, Darmstadt, Germany). Anti-rabbit (sc-2547, Santa Cruz Biotechnology, Dallas, TX, USA) was used as a secondary antibody.

### 2.7. Statistical Analysis

The data were presented as means ± SEM. A two-way ANOVA was used to assess the significance of the effects of the BLM treatment and the LR-MSC donor in the in vivo experiment and the effects of the TGFβ treatment and the LR-MSC donor in the in vitro experiment. A Bonferroni post hoc test was conducted for multiple comparisons within groups. Statistical analysis was performed using GraphPad Prism 8.4.0 (GraphPad Software, Inc., Boston, MA, USA) software. The threshold of significance was set at *p* < 0.05.

## 3. Results

### 3.1. Instillation of Either IPF or CON LR-MSC Sustained BLM-Induced Histological and Cellular Signs of Lung Damage

Mice survived until the end of the study protocol, except for one mouse belonging to the BLM + IPF group, who died on d12. As expected, BLM administration induced a significant body weight reduction from day 2 onwards in the BLM + VEH mice (*p* < 0.001), which was the highest on d4 (12.6% reduction compared to d0) and persisted on d14 (10.4%). To analyze the effect of LR-MSC administration, mouse body weights on days 4, 8, 10, and 14 were compared to the day of cell instillation (d2) (Table 1). Within PBS-treated mice, those instilled with IPF LR-MSC showed a significantly lower body weight than PBS + VEH (*p* < 0.05). Within BLM-treated mice, those instilled with CON LR-MSC had a significantly higher body weight than BLM-VEH (*p* < 0.05), whereas there were no significant differences between BLM + VEH and BLM + IPF mice. These data indicated that mice receiving IPF LR-MSC exhibited worse maintenance of their body weight.

The quantification of lung damage using the modified Ashcroft scale showed the BLM-induced development of lung fibrosis (*p* < 0.001), which was not different between the experimental groups of mice treated with BLM (Figure 1B). The appearance of lung fibrosis followed a heterogeneous pattern and, on average, was classified as moderate fibrosis (Appendix A–F). In those fields in which fibrosis was noticeable, this was loose and incipient (tiny interstitial aggregate of spindle-shaped fibroblasts in a pale-staining myxoid background), and no mature eosinophilic collagen fibrotic depots were observed (Appendix A–F). Incipient fibrosis was accompanied by signs of interstitial inflammation, and both events were observed in 4/7 BLM + VEH, 4/7 BLM + CON, and 2/5 BLM + IPF mice (Appendix A–F). In PBS-treated mice, signs of inflammation without fibrosis were observed in 1/5 PBS + CON and 1/5 PBS + IPF mice, and in any mouse in the PBS + VEH group (Appendix A–C). According to the average Ashcroft score of the 10 lung fields, BLM-induced lung damage was only significant in BLM + VEH and BLM + CON mice compared with their controls, and there were no significant differences between mice receiving IPF and CON LR-MSC, neither in BLM- nor in PBS-treated mice (Figure 1B). The quantitative assessment of collagen accumulation by picrosirius red staining revealed that there were no significant differences in response to BLM or LR-MSC administration (Figure 1C and Appendix A), in accordance with the observation of a lack of mature eosinophilic collagen fibrotic depots.

The BALF immune cell populations estimation showed an increase in the number of lymphocytes (*p* < 0.01), macrophages (*p* < 0.005), and granulocytes (*p* < 0.001) induced by BLM instillation (Figure 1D–F). The number of BLM-induced lymphocytes and macrophages was not different between the BLM groups, while in granulocytes, the effect of BLM was only significant in the BLM + CON mice (*p* < 0.05). In contrast to the BLM effect, LR-MSC instillation did not affect the number of immune cell populations.

### 3.2. IPF or CON LR-MSC Instillation Did Not Affect BLM-Induced Expression of Markers of Lung Damage

The mRNA expression levels of genes coding for extracellular matrix proteins (*Col1a1* and *Fn*) and proinflammatory cytokines (*Il-6*, *Tnfα*, *Mcp1*, and *Il-1β*) were analyzed in lung tissues. As expected, BLM administration significantly induced both *Col1a1* (*p* < 0.05) and *Fn1* (*p* < 0.01) mRNA expression levels (Figure 2A,B). Within BLM-treated mice, there were no significant differences in the expression of these genes, although *Col1a1* induction was more evident in the BLM + VEH group (*p* = 0.06). Likewise, BLM treatment significantly induced the *Il6* mRNA expression levels (*p* < 0.05) (Figure 2C), whereas the expression of the other genes involved in inflammation was not changed by BLM (Figure 2D–F). Within BLM-treated mice, there were no significant differences in the expression of *Il6* mRNA, although the induction was more evident in the BLM + IPF group (*p* = 0.101).

The protein levels of markers of myofibroblast (aSMA), epithelial cell remodeling (KRT8), and epithelial apoptosis (KRT18) were analyzed in lung tissues. As expected, BLM administration induced the aSMA protein levels (*p* < 0.05) (Figure 2G,H). An increased number of KRT8-positive cells is a feature of IPF and is increased by repetitive BLM administration [31,32,33]. In the present study, BLM administration failed to significantly induce the KRT8 and KRT18 protein levels (Figure 2G,I,J). However, a significantly different cell donor effect was observed in the KRT8 protein levels (*p* < 0.01) (Figure 2I), and the same tendency was noted in the KRT18 protein levels (*p* = 0.06) (Figure 2J). To understand the effect of LR-MSC on these proteins, the KRT8/KRT18 ratio was calculated. Despite a tendency towards decreasing KRT18 levels in BLM + IPF mice, no significant differences were obtained regarding the KRT8/KRT18 ratio between BLM-treated groups (Figure 2K).

### 3.3. Lower TGFβ Response in the Profibrotic Gene Expression in IPF vs. CON LR-MSC

First, we analyzed aSMA and FN1 protein levels in isolated and cultured LR-MSC treated with or without 10 ng/mL TGFβ for 48 h. IPF LR-MSC displayed robust basal aSMA protein expression, and as expected, their levels were significantly higher than those found in CON cells (*p* < 0.01) (Figure 3A,C). Likewise, FN1 protein expression levels were significantly higher in IPF vs. CON LR-MSC (*p* < 0.005) (Figure 3B,C). TGFβ treatment failed to significantly increase aSMA protein levels, as they were increased by 3.1-fold in CON cells but only by 1.3-fold in IPF cells. On the other hand, TGFβ treatment increased FN1 protein levels in CON (1.7-fold) and IPF (1.3-fold) cells, an effect that was statistically significant (*p* < 0.01).

Then, the TGFβ response was further explored by treating IPF and CON LR-MSC to different TGFβ concentrations (1, 10, or 20 ng/mL for 24 h). We analyzed the effect of TGFβ treatment on the mRNA levels of TGFb itself, ACTA2 (myofibroblast marker), COL1A1, FN1, and TNC (extracellular matrix components). TGFβ treatment significantly induced TGFb expression (*p* < 0.005), but this effect was only significant in the CON LR-MSC exposed to the highest TGFβ concentration (Figure 3A). mRNA levels of ACTA2 in response to TGFβ were significantly greater in CON LR-MSC cells than in IPF cells (*p* < 0.01) (Figure 3B). In line with the response on ACTA2, the TGFβ effect on COL1A1, FN, and TNC gene expression was significantly greater in CON LR-MSC cells than in IPF cells (*p* < 0.05 in all cases) (Figure 3C–E).

## 4. Discussion

Growing evidence supports that LR-MSC are a major source of fibroblasts and myofibroblasts in IPF and, hence, LR-MSC would actively contribute to fibrosis [12,13,14,15,20,21]. Also, mitochondrial dysfunction has been described in IPF LR-MSC [19], and other molecular signs of aging, including stem cell exhaustion, are also expected to be present in these cells [2,7]. Thus, LR-MSC would “passively” contribute to fibrosis by limiting their potential to regenerate the damaged epithelium. Both mechanisms are likely involved in IPF progression, though the responsiveness of LR-MSC from IPF patients to profibrotic stimuli is still unclear and remains to be elucidated as to how they participate in IPF progression. In the present study, to better understand the role of LR-MSC in the progression of pulmonary fibrosis, we analyzed the behavior of both IPF and CON LR-MSC in vivo and in vitro after being exposed to profibrotic stimuli. In BLM-treated mice, the lung effects triggered by LR-MSC from both cohorts were not different from BLM + VEH mice. In contrast, the in vitro response to TGFβ on the expression of fibroblast and myofibroblast markers was clearly attenuated in IPF LR-MSC, suggesting that these cells may have partly lost their responsiveness to profibrotic stimuli.

Numerous studies have shown the capacity of MSC to reduce BLM-induced lung fibrosis and inflammation, which prompted them to prove their therapeutic potential in clinical trials [34,35]. Later, the potential of MSC to treat IPF was focused on the use of MSC-derived extracellular vesicles [36]. In the BLM model, such an effect is influenced by experimental factors, as discussed in [37,38]. In our study, however, CON LR-MSC did not reduce lung BLM-induced fibrosis, probably because the cells were obtained from elderly matched individuals, which may negatively affect the MSC capabilities [39]. A beneficial effect was not triggered by IPF LR-MSC either, as could be expected from previous results [40], in which bone marrow MSC (BM-MSC) from IPF patients did not prevent the evolution of lung fibrosis in BLM-treated mice. Interestingly, the administration of BM-MSC [40] and LR-MSC (present study) from IPF patients to BLM-treated mice affected body weight regain, suggesting that MSC from IPF patients release some factors that may inhibit lung repair and, in addition, affect other organs systemically. These LR-MSC-derived factors might include Il-6 and Il-1b, pro-inflammatory cytokines involved in body weight maintenance [41,42], that are overexpressed in IPF LR-MSC [23], and sustained or increased in the lungs of BLM-treated mice irrespective of using BM- [40] or LR-MSC (present study) from IPF patients. Other LR-MSC-derived factors that could be involved in the defective lung repair may include exosomes and other types of extracellular vesicles, since it has been demonstrated that their production from other IPF lung cells contributes to disease progression [43,44,45,46,47,48,49], and owing to the anti-fibrotic activity of MSC-derived extracellular vesicles [36,50]

Our hypothesis was that the effect of LR-MSC administration on BLM-induced lung damage in mice may depend on whether the donor is an IPF patient or not. This hypothesis is supported by recent evidence showing that the features and function of LR-MSC and progenitor cells are altered in IPF, which would explain their potential contribution to fibrosis. In particular, in vitro studies have shown that IPF lung-resident progenitor cells tend to differentiate into myofibroblasts [11,20], express higher levels of inflammatory-related genes, induce a pathological phenotype in surrounding MSC and fibroblasts [23], and exhibit mitochondrial dysfunction [19]. Then, it could be expected that the IPF environment induces LR-MSC to exacerbate lung fibrosis. However, our in vivo results did not support it, and several reasons could explain it. First, there is the ratio between the degree of lung damage and the number of instilled cells, as the acute lung damage could be too big to observe any additional effect mediated by LR-MSC, and/or because the number of cells was too low. Second is the communication between murine and IPF patient cells, because BLM-exposed murine cells might be poorly responsive to the factors involved in promoting fibrosis released by the LR-MSC from IPF patients and/or, vice versa, the IPF LR-MSC might be poorly responsive to the murine BLM-induced profibrotic microenvironment. The latter agrees with our in vitro results, which indicated that IPF LR-MSC are indeed less responsive than CON cells to TFGβ, suggesting that LR-MSC from IPF patients has mitigated their ability to acquire a profibrotic phenotype and to promote, at least actively, lung fibrosis.

By contrast, in the initial stage of pulmonary fibrosis, it is thought that LR-MSC are activated and continuously stimulated to proliferate and differentiate into myofibroblasts [7]. In our study, we showed that IPF LR-MSC displayed robust protein expression of the myofibroblast marker aSMA in basal conditions, in agreement with other studies [23]. The aSMA expression indicates that LR-MSC had been transformed because of their previous exposure to a profibrotic environment in the human lung. Their responsiveness to TGFβ might be attenuated at a certain point of IPF progression. Thus, differences in the features and behavior of LR-MSC seem to occur throughout IPF evolution. Supporting this argument, Chanda et al. [51] demonstrated that BAL-derived mesenchymal stromal cells from progressive IPF patients have reduced FGF-10 expression compared to cells from stable IPF patients, suggesting that, in an advanced disease stage, LR-MSC have lost the potential to regenerate the damaged epithelium.

## 5. Conclusions

Previous evidence, together with our results, allows us to conclude that IPF LR-MSC differentiate into myofibroblasts; thereby, they actively contribute to IPF. However, the IPF LR-MSC exhaust and the rest of the LR-MSC display a particular phenotype. They are transformed into cells expressing aSMA with an apparent decrease in their responsiveness to TGFβ to further increase the expression of ECM components and the myofibroblast marker. At the same time, they have acquired senescence-related factors, which could explain why they do not reduce the induced lung fibrosis and may contribute to the limited repair ability. However, functional studies investigating the relationships between LR-MSC from IPF patients and damaged epithelial cells are required to improve our understanding of the role of LR-MSC in the IPF.

## Figures and Tables

**Figure 1 cells-13-00160-f001:**
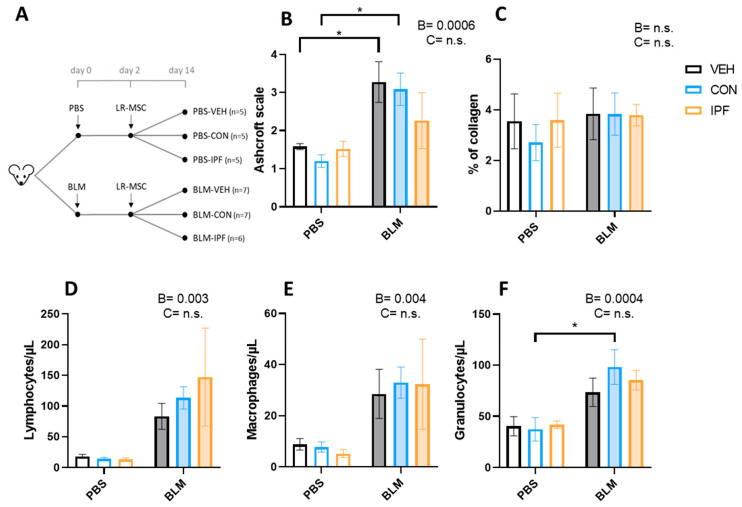
LR-MSC sustained BLM-induced histological and cellular signs of lung damage. (**A**) Experimental design. (**B**) Lung damage assessed using the modified Ashcroft score. (**C**) Lung collagen quantification assessed with Picrosirius Red staining. Quantification of lymphocytes (**D**), macrophages (**E**), and granulocytes (**F**) via flow cytometry in the bronchoalveolar lavage fluid. Data are expressed as means ± SEM, *n* = 5–7. Two-way ANOVA, B: BLM effect, C: cell donor effect; and a *p* value is indicated for each factor if the effect is statistically significant (*p* < 0.05); n.s., not significant. Bonferroni post hoc: * *p* < 0.05 vs. PBS groups. PBS: phosphate-buffered saline; BLM: bleomycin; LR-MSC: lung-resident mesenchymal stem cells; VEH: vehicle; CON: control; and IPF: idiopathic pulmonary fibrosis.

**Figure 2 cells-13-00160-f002:**
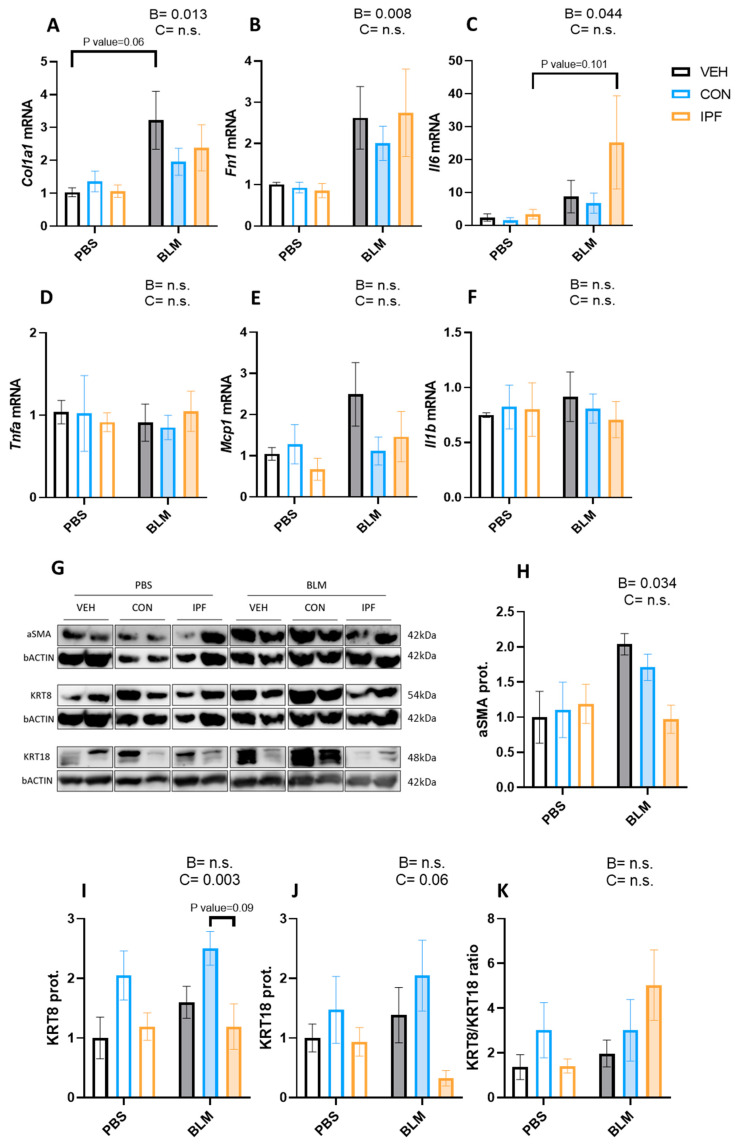
LR-MSC sustained BLM-induced molecular signs of lung damage. mRNA expression levels of collagen (**A**), fibronectin 1 (**B**), interleukin 6 (**C**), tumor necrosis factor alpha (**D**), monocyte chemoattractant protein 1 (**E**), and interleukin 1 beta (**F**). Glyceraldehyde-3-phosphate dehydrogenase and hypoxanthine phosphoribosyltransferase 1 were used as housekeeping genes. Two representative bands of Western blot for each experimental group were selected to show protein levels (**G**). Protein levels of alpha smooth muscle actin (**H**), cytokeratin 8 (**I**), and cytokeratin 18 (**J**). Beta-actin levels were used as a loading control. Cytokeratin 8 to cytokeratin 18 ratio (**K**). Data are expressed as means ± SEM, *n* = 5–7. Two-way ANOVA, B: BLM effect, C: cell donor effect; a *p* value is indicated for each factor if the effect is statistically significant (*p* < 0.05); n.s., not significant. Bonferroni post hoc. PBS: phosphate-buffered saline; BLM: bleomycin; LR-MSC: lung-resident mesenchymal stem cells; VEH: vehicle; CON: control; and IPF: idiopathic pulmonary fibrosis.

**Figure 3 cells-13-00160-f003:**
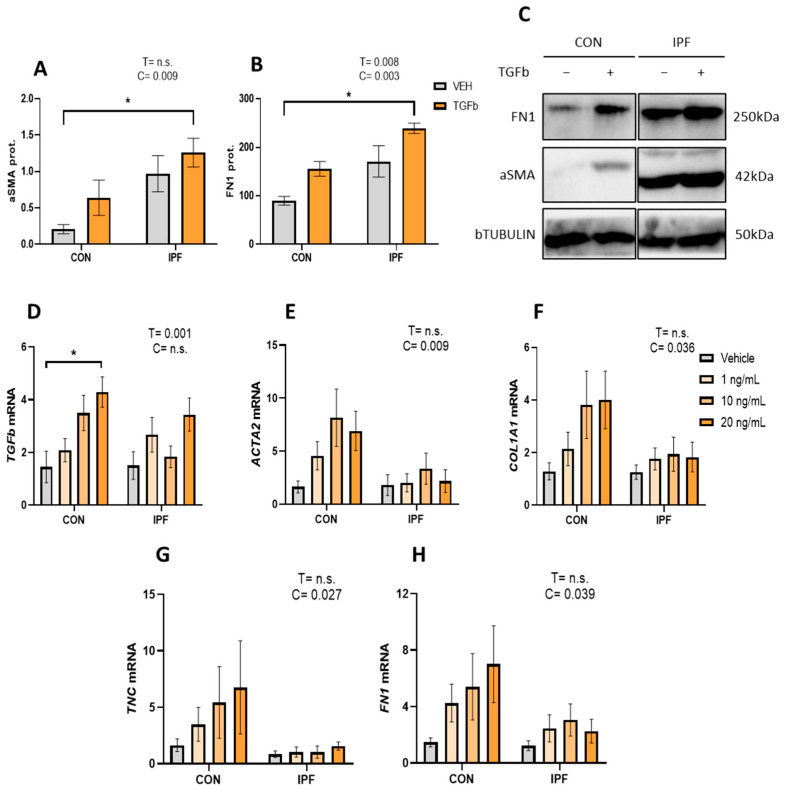
Attenuated TGFβ response in IPF vs. CON LR-MSC. Protein expression levels of alpha smooth muscle actin (**A**) and fibronectin (**B**). Beta tubulin levels were used as a loading control. Representative bands of the Western blot show protein expression levels (**C**). mRNA expression levels of transforming growth factor beta (**D**), actin alpha 2 (**E**), collagen (**F**), tenascin C (**G**), and fibronectin 1 (**H**). Beta 2 microglobulin was used as a housekeeping gene. Data are expressed as means ± SEM, *n* = 3 (**A**,**B**), and *n* = 7 (**D**–**H**). Two-way ANOVA, T: TGBβ effect, C: cell donor effect; a *p* value is indicated for each factor if the effect is statistically significant (*p* < 0.05); n.s., not significant. Bonferroni post hoc: * *p* < 0.05 vs. LR-MSC treated with vehicle. LR-MSC: lung-resident mesenchymal stem cells; CON: control; and IPF: idiopathic pulmonary fibrosis.

**Table 1 cells-13-00160-t001:** Worse body weight maintenance in mice instilled with IPF LR-MSC.

Day	PBS + VEH	PBS + CON	PBS + IPF	BLM + VEH	BLM + CON	BLM + IPF	*p* Value		
							B	C	B × C
4	101.2 ± 1.5	98.5 ± 1.4	98.9 ± 1.2	95.1 ± 2.2 *	97.7 ± 0.9	95.8 ± 2.0	0.021	n.s.	n.s.
8	105.7 ± 1.9	99.6 ± 1.6	101.7 ± 1.4	95.3 ± 2.7 *	101.4 ± 2.0	92.5 ± 3.8 #	0.007	n.s.	0.04
10	105.1 ± 1.9	101.4 ± 2.1	102.5 ± 2.0	95.3 ± 3.4	98.9 ± 1.9	93.4 ± 4.0	0.005	n.s.	n.s.
14	108.2 ± 1.5	103.1 ± 2.1	100.7 ± 1.6 *	97.4 ± 2.6 *	104.2 ± 0.9 °	99.0 ± 0.7	0.015	n.s.	0.01

Body weight percentages relative to day 2. Mice were treated with bleomycin or PBS on day 0 and received LR-MSC on day 2. Data were expressed as means ± SEM, *n* = 5–7. Two-way ANOVA, B: BLM effect, C: cell donor effect, B × C: interaction effect, and n.s., not significant. Bonferroni post hoc: * *p* < 0.05 vs. PBS + VEH; # *p* < 0.05 vs. BLM-CON; and ° *p* < 0.05 vs. BLM + VEH.

## Data Availability

Data are contained within the article and Appendix A.

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
