# Peer review of "In Vivo and In Vitro Pro-Fibrotic Response of Lung-Resident Mesenchymal Stem Cells from Patients with Idiopathic Pulmonary Fibrosis"

_cells, 2024, doi:10.3390/cells13020160_

Round 1
Reviewer 1 Report
Comments and Suggestions for Authors
Major comments:
In the result section, 3.1 the authors report Ashcroft scaling as the following:
“Ashcroft scale showed the BLM-induced development of lung fibrosis (p<0.001), which was not different between the experimental groups of mice treated with BLM (Figure 1B). The appearance of lung fibrosis followed a heterogeneous pattern and, on average, was classified as moderate fibrosis. In those fields in which fibrosis was noticeable, this was loose and incipient (tiny interstitial aggregate of spindle-shaped fibroblasts in a pale-staining myxoid background), and no mature eosinophilic collagen fibrotic depots were observed (not shown). “
Given the nature of Ashcroft scale, I believe some representative images are needed to support the claim. I recommend inserting some representative images for figure 1C as well.
Why are there two bands for each target protein expression in figure 2G? Do they come from different animals? Also, the bands from BLM + IPF samples did not show an obvious increase in level.
In result section 3.3, figure 3A the authors claimed that aSMA was not significantly increased by the TGF-beta treatment. But the increase in CON group looks like should be significant. Can the authors double check the statistics here?
Any specific reason of why two regimes of TGF-beta treatment were tested? 24hr VS 48 hr, and different markers were analyzed in each of the regime.
Have the authors considered the cross-species administration of MSCs? How does the study represent the contribution of MSC in IPF if they are tested in a different species?
Minor comments:
I believe “CON” is reference MSC from control group patient, but it wasn’t defined anywhere in the abstract.
Figure 2K was mis-labelled in the figure legend.
Comments on the Quality of English LanguageN/A
Reviewer 2 Report
Comments and Suggestions for Authors
In a field that most research in pulmonary fibrosis is evaluated in interactions between tissue resident alveolar macrophages, AT Cells and fibroblasts, this was an interesting paper evaluating the lung resident MSCs and their effects on fibrosis. The authors evaluated both in vitro and in vivo effects.
1) It would be interesting to see the level of disease in IPF (Stage I vs Stage III) and how that affects the response to fibrosis.
2) It is interesting that the cells isolated were validated using a Stemflow kit. Do lung-resident MSCs behave differently and/or are they phenotypically different from MSCs?
3) Do MSC-derived exosomes in disease format differ in any way? This might be an interesting finding.
4) Do the IPF LR-MSCs transdifferentiate and is there any evidence to suggest this?
5) What is the role of the resident immune cell population in this? (Eg tissue resident macrophages).
Reviewer 3 Report
Comments and Suggestions for Authors
1. Please use immunofluorescence staining or flow cytometry data to show patient LR-MSCs are actually engrafted in B6 mouse. I suspect that they are rejected by the immune system.
2. Please clarify how statistical analysis was done. It is very confusing by the way that authors presented.
Round 2
Reviewer 1 Report
Comments and Suggestions for Authors
The authors have addressed my comments on the original submission except the for following:
I appreciate the authors include some representative images in the revised submission. However, the quality of Figure S2 is not good enough to represent the author’s claim. The images provided are all very dark in color tone. The collagen content does not stand out.
My previous comment on figure 2G was talking about why two separate bands were reported side-by-side. I was not talking about there are double bands in a one single crop. Can the authors address this again?
Thank you.
Reviewer 3 Report
Comments and Suggestions for Authors
1. If the cells cannot be detected, how do they regulate/affect fibrosis in mice.
2. As you pointed out, MSCs may release exosomes which contribute to disease progression. If this is, can you use exosome inhibitors to rescue the phenotype?
